TOPICAL REVIEW

# The synaptic vesicle cluster as a controller of pre- and postsynaptic structure and function

Sofiia Reshetniak[1] , Cristian A. Bogaciu[1] , Stefan Bonn[2] , Nils Brose[3,26], Benjamin H. Cooper[3], Elisa D'Este[4], Michael Fauth[5], Rubén Fernández-Busnadiego[6,26] , Maksims Fiosins[7], André Fischer[8,26] , Svilen V. Georgiev[1], Stefan Jakobs[9,26], Stefan Klumpp[10] , Sarah Köster[11,26], Felix Lange[9] , Noa Lipstein[7] , Victor Macarrón-Palacios[4] , Dragomir Milovanovic[12] , Tobias Moser[13,26] , Marcus Müller[14] , Felipe Opazo[1] , Tiago F. Outeiro[15], Constantin Pape[16], Viola Priesemann[5,17,26], Peter Rehling[18,26] , Tim Salditt[11,26], Oliver Schlüter[19], Nadja Simeth[20,26] , Claudia Steinem[20,26], Tatjana Tchumatchenko[21], Christian Tetzlaff[1], Marilyn Tirard[3] , Henning Urlaub[22], Carolin Wichmann[23,24], Fred Wolf[25] and Silvio O. Rizzoli[1,26]

Handling Editors: Laura Bennet & Samuel Young

The peer review history is available in the Supporting Information section of this article (https://doi.org/10.1113/JP286400#support-information-section).

**Abstract figure legend** The synaptic vesicle cluster acts as a master regulator of synaptic structure and function, through various involvements in processes that range from synaptic initiation to plasticity and synaptic loss.

*The Journal of Physiology*

**Abstract** The synaptic vesicle cluster (SVC) is an essential component of chemical synapses, which provides neurotransmitter-loaded vesicles during synaptic activity, at the same time as also controlling the local concentrations of numerous exo- and endocytosis cofactors. In addition, the SVC hosts molecules that participate in other aspects of synaptic function, from cytoskeletal components to adhesion proteins, and affects the location and function of organelles such as mitochondria and the endoplasmic reticulum. We argue here that these features extend the functional involvement of the SVC in synapse formation, signalling and plasticity, as well as synapse stabilization and metabolism. We also propose that changes in the size of the SVC coalesce with changes in the postsynaptic compartment, supporting the interplay between pre- and postsynaptic dynamics. Thereby, the SVC could be seen as an 'all-in-one' regulator of synaptic structure and function, which should be investigated in more detail, to reveal molecular mechanisms that control synaptic function and heterogeneity.

(Received 12 June 2024; accepted after revision 11 September 2024; first published online 3 October 2024)

**Corresponding authors** S. Reshetniak: Institute for Neuro- and Sensory Physiology and Biostructural Imaging of Neurodegeneration (BIN) Centre, University Medical Centre Göttingen, Göttingen 37 073, Germany. Email: sofiia.reshetniak@med.uni-goettingen.de

S. O. Rizzoli: Institute for Neuro- and Sensory Physiology and Biostructural Imaging of Neurodegeneration (BIN) Centre, University Medical Centre Göttingen, Göttingen 37 073, Germany. Email: srizzol@gwdg.de

[1] *Institute for Neuro- and Sensory Physiology and Biostructural Imaging of Neurodegeneration (BIN) Center, University Medical Center Göttingen, Göttingen, Germany*

[2] *Institute of Medical Systems Biology, Center for Molecular Neurobiology Hamburg, Hamburg, Germany*

[3] *Department of Molecular Neurobiology, Max Planck Institute for Multidisciplinary Sciences, Göttingen, Germany*

[4] *Optical Microscopy Facility, Max Planck Institute for Medical Research, Heidelberg, Germany*

[5] *Georg-August-University Göttingen, Faculty of Physics, Institute for the Dynamics of Complex Systems, Friedrich-Hund-Platz 1, Göttingen, Germany*

[6] *Institute of Neuropathology, University Medical Center Göttingen, Göttingen, Germany*

[7] *Leibniz-Forschungsinstitut für Molekulare Pharmakologie, Berlin, Germany*

[8] *German Center for Neurodegenerative Diseases, Göttingen, Germany*

[9] *Research Group Structure and Dynamics of Mitochondria, Max Planck Institute for Multidisciplinary Sciences, Göttingen, Germany*

[10] *Theoretical Biophysics Group, Institute for the Dynamics of Complex Systems, Georg-August University Göttingen, Göttingen, Germany*

[11] *Institute for X-Ray Physics, Georg-August University Göttingen, Göttingen, Germany*

[12] *Laboratory of Molecular Neuroscience, German Center for Neurodegenerative Diseases, Berlin, Germany*

[13] *Institute for Auditory Neuroscience, University Medical Center Göttingen, Göttingen, Germany*

[14] *Institute for Theoretical Physics, Georg-August University Göttingen, Göttingen, Germany*

[15] *Department of Experimental Neurodegeneration, Center for Biostructural Imaging of Neurodegeneration, University Medical Center Göttingen, Göttingen, Germany*

[16] *Institute of Computer Science, Georg-August University Göttingen, Göttingen, Germany*

[17] *Max-Planck Institute for Dynamics and Self-Organization, Am Fassberg 17, Göttingen, Germany*

[18] *Department of Cellular Biochemistry, University Medical Center Göttingen, Göttingen, Germany*

[19] *Clinic for Psychiatry and Psychotherapy, University Medical Center Göttingen, Göttingen, Germany*

[20] *Institute of Organic and Biomolecular Chemistry, Georg-August University Göttingen, Göttingen, Germany*

[21] *Institute of Experimental Epileptology and Cognition Research, University of Bonn Medical Center, Bonn, Germany*

[22] *Bioanalytical Mass Spectrometry, Max Planck Institute for Multidisciplinary Sciences, Göttingen, Germany*

[23] *Institute for Auditory Neuroscience University Medical Center Göttingen, Göttingen, Germany*

[24] *Biostructural Imaging of Neurodegeneration (BIN) Center, University Medical Center Göttingen, Göttingen, Germany*

Synapses are the central information processors in the brain. Their function, efficacy and plasticity are key determinants of all brain functions, as well as of the corresponding behavioural output. Conversely, aberrant synapse function is the cause of many neurological and psychiatric disorders. Our consortium, SFB1286 'Quantitative Synaptology,' was initiated in 2017, aiming to analyze the molecular details of synapses, and generate a model able to reproduce synaptic function and plasticity. We started by determining the molecular composition of the synapse, and its changes during synaptic activity, and complemented this with several key functional parameters, ranging from synaptic protein mobility to the principles behind the formation of synaptic active zones. The structural and functional parameters are currently being integrated in modelling studies aiming to explain and predict synaptic function and plasticity, as well as to test the hypothesis that specific synaptic diseases are caused by differences in the numbers, activity and/or organization of defined synaptic elements.

[25] *Max-Planck-Institute for Dynamics and Self-Organization, 37077 Göttingen and Institute for Dynamics of Biological Networks, Georg-August University Göttingen, Göttingen, Germany*

[26] *Cluster of Excellence "Multiscale Bioimaging: from Molecular Machines to Networks of Excitable Cells" (MBExC), University of Göttingen, Göttingen, Germany*

## Introduction

Chemical synapses show an enormous structural and functional diversity (Nusser, 2018; Wichmann & Kuner, 2022; Zhai & Bellen, 2004). This diversity occurs on different levels, starting from structural and functional features of the presynaptic terminal, as well as the post-synaptic partner. The structural heterogeneity of pre-synapses ranges from simple bouton-like terminals as found for hippocampal Schaffer collateral–CA1 contacts (Dürst et al., 2022; Lee et al., 2020; Maus et al., 2020) to huge presynaptic endings harboring hundreds of individual active zones (AZs), as for calyceal auditory brain stem synapses (Dondzillo et al., 2010; Schneggenburger & Forsythe, 2006) or mossy fibre–CA1 contacts (Lee et al., 2020; Maus et al., 2020; Zhao et al., 2012). Additionally, synapses can exhibit different functional properties, such as differing release probabilities, or being facilitating or depressing. Finally, synapses might contain highly specialized structures, such as the synaptic ribbon that is characteristic to many sensory synapses (Schmitz, 2009).

Nonetheless, all chemical synapses tend to consist of the same set of modules (Wichmann & Kuner 2022; Zhai & Bellen, 2004): the presynaptic terminal with the AZ, which forms a scaffold that organizes the synaptic vesicle (SV) recruitment and release, the synaptic vesicle cluster (SVC), the synaptic cleft, and the postsynaptic compartment. Thus, all chemical synapses follow a common principle not only structurally, but also functionally, to initiate neurotransmission: SVs need to be recruited from the SVC to the AZ, become tightly primed to the membrane, and will then fuse and release neurotransmitter.

The SVCs typically contain a few hundred to a few thousand vesicles, and support neurotransmitter release by maintaining a pool of release-ready SVs, which can be consumed gradually by prolonged rounds of high-frequency stimulation (Birks et al., 1960; Heuser & Reese, 1973). Following fusion, the vesicles are retrieved by endocytosis, and are prepared for new rounds of release (Chanaday et al., 2019; Gauthier-Kemper et al., 2015; Rizzoli, 2014). The number of vesicles actively participating in exocytosis at any point in time is relatively low under physiological activity rates, at least for large synapses (Denker et al., 2011a, 2011b). In principle, a handful of vesicles would be sufficient to ensure neuro-transmission for long time periods, even at smaller synapses (Rose et al., 2013) because the release probability is relatively low (10–30%) (Gandhi & Stevens, 2003), the recycling process lasts only a few tens of seconds and the overall spike rate can be lower than 5 Hz even during active behaviours (Hirase et al., 2001).

Nevertheless, synapses contain, as mentioned above, hundreds to thousands of vesicles, and not just the few (<100 for average synapses) that are immediately required for neurotransmission. This was recognized early, with many works separating vesicles into an actively recycling pool and a 'reserve pool' (Birks et al., 1960; Elmqvist & Quastel, 1965; Liley & North, 1953). Their suggested potential roles varied over the decades, from (1) a pool of vesicles that ensures the rapid refilling of the recycling pool during emergencies, or (2) as a neurotransmitter storage compartment, to (3) a system buffering components for the SV recycling, such as end-ocytosis cofactors (cofactor buffering hypothesis) (Denker et al., 2011a, 2011b; Shupliakov, 2009). The first two hypotheses probably do not explain the general need for a large reserve vesicle pool, since, as mentioned above, only a few SVs are typically released per activity burst, and little exchange of neurotransmitter between the vesicle pools is known to take place (Van der Kloot, 2003). Accumulated evidence demonstrates that the third view is substantially more valid, with vesicles indeed collecting various soluble molecules, and providing them to the rest of the synapse during activity (Bai et al., 2010; Denker et al., 2011b; Reshetniak et al., 2020; Shupliakov, 2009). This does not mean that other roles should be completely discounted, however, especially because small synapses may actually use their (relatively smaller) reserve pools regularly (Rose et al., 2013). Nevertheless, the main role of most of the vesicles in the SVC appears to be that of keeping cofactor proteins in the synapse.

Modern views have focused on the process in which these vesicles and cofactors assemble and accumulate, leading to the current view of their organization by liquid-liquid phase separation (LLPS). In cell biology, LLPS involves the segregation of molecules into a condensed assembly with defined boundaries, which can be reversed in response to specific physiological signals (as opposed to irreversible aggregation), leading to the formation of two or more distinct phases with differing concentrations of components. Within cellular environments, this phenomenon gives rise to so-called membrane-less compartments, which are now considered organelles, and is facilitated by numerous protein–protein interactions, often mediated by intrinsically disordered domains (Alberti, 2017; Shin & Brangwynne, 2017). The main advantage of LLPS

is that exchange between the two phases is relatively easy, compared to membrane-dependent separation, or aggregates. It has been originally suggested that the SVC is formed via LLPS by the interaction of synapsin and SVs (Milovanovic & Camilli, 2017; Milovanovic et al., 2018), with the intrinsically disordered region of synapsin being critical for this process (Hoffmann et al., 2023; Pechstein et al., 2020). In the following years, growing evidence demonstrated that not only synapsin, but also other proteins, including soluble trafficking cofactors (discussed in detail below), take part in this process (Hoffmann et al., 2021; Jackson et al., 2024; Park et al., 2021; Ray et al., 2020; Sigrist & Haucke, 2023; Yoshida et al., 2023).

Overall, the modern LLPS view is in good agreement with the cofactor buffering hypothesis. However, one other issue complicates this view: the SV proteome contains much more than *bona fide* SV proteins or cofactors (O'Neil et al., 2021; Takamori et al., 2006; Taoufiq et al., 2020; Wilhelm et al., 2014). Indeed, the current estimated range is for between 400 and 1400 proteins being found within the SVC, albeit not all synapse subtypes would contain all of these proteins (Fig. 1 . Many such molecules may be present in SVCs only as traces or contaminants, or may reside on special populations of vesicles (Binotti et al., 2024), but some have long been demonstrated to interact with vesicles and play true physiological roles, independent of SV recycling.

Many key synaptic proteins are found at low levels in synapses (below 100 copies), including several scaffolds, adhesion molecules or receptors (Wilhelm et al., 2014). The recruitment of such molecules by the SVC would affect their local abundances and relative concentrations, and could also generate functional nanodomains. This implies that the SVC could do much more than provide vesicles for release and buffer the cofactors needed for their recycling: it could serve as a hub for molecular processes that shape completely different synaptic properties. We suggest here that it does so, and that it may control surprisingly many aspects of synaptic physiology, on both sides of the synapse, thus providing a new view on the potential role of the SVC. Throughout this review, we discuss different processes in which the SVC could influence synaptic structure and function, from the effects of the SVC on synaptogenesis and synaptic organelle organization, to its role in plasticity and synaptic turnover, on both sides of the synapse, as summarized in Fig. 2. While some of the claims discussed here are well established in the field, others are only our speculations, based on the available data, and are open to further discussion and, eventually, experimental confirmation (Table 1).

Here, we keep our view simple, by focusing on synapses with one SVC, with a modular, coherent function, and one AZ. We are confident that further research will determine inhomogeneities in SVCs as a result of the presence of heterogeneous SVs in different locations (e.g. old or young SVs, or SVs with different molecular markers), leading to the formation of subdomains within single synapses, which participate differently in neurotransmitter release (Guzikowski & Kavalali, 2021), or serve different plasticity roles (Schlüter et al., 2006). Nonetheless, because few details are available on the organization of such heterogenous subdomains, we do not discuss them here, but rather focus on an idealized, functionally homogenous SVC.

## Synapse initiation

It is generally assumed that neurons have cell-autonomous genetic programs to develop axons, presumably encoded by their epigenomes. Axons form even in isolated neurons, in culture, in the absence of the organized environment of the brain, and will search for sites where they can produce synapses (Barnes & Polleux, 2009; Dotti et al., 1988). It has been known for decades that growing axons also release neurotransmitters, and that they can form synapse-like arrangements within minutes (or even seconds) of meeting a suitable target, albeit most of these contacts will be lost, with only a few becoming true synapses, in connection with external cues (Polleux & Snider, 2010). Adhesion molecules are critical to this process, with their loss changing the pattern of synapse formation and/or the functionality of the resulting synapses (Südhof, 2021).

Even before axon formation, neurons begin to exhibit some degree of polarization, by accumulating membrane organelles and increasing vesicular transport in one of the neurites (Bradke & Dotti, 1997). The increased membrane flow ensures that the axon has the necessary vesicles for the continuous membrane enlargement (Kraszewski et al., 1995). The need for substantial amounts of membranes for axon growth, and the fact that a localized and specialized recycling machinery for these membranes does not exist, means that a mixed type of vesicles will be used, generated by the typical endosomal machinery for membrane processing (Pasterkamp & Burk, 2021; Rizalar et al., 2023), without exhibiting substantial synaptic specialization. More information on axonal proteomes is provided by Poulopoulos et al. (2019). Nonetheless, the axon is only programmed to transport presynaptic-like cargoes, implying that the vesicles used for the membrane flow that builds the initial axons will also contain most, if not all, SV proteins. Although perturbations of SV and AZ proteins delivery to the synapse do not lead to defects in axon growth (Marcó de la Cruz et al., 2024; Rizalar et al., 2023; Vukoja et al., 2018), we speculate that this mixed population of non-specialized vesicles are driving both growth of the axon and eventual synapse formation. Experimental evidence corroborates this idea because an

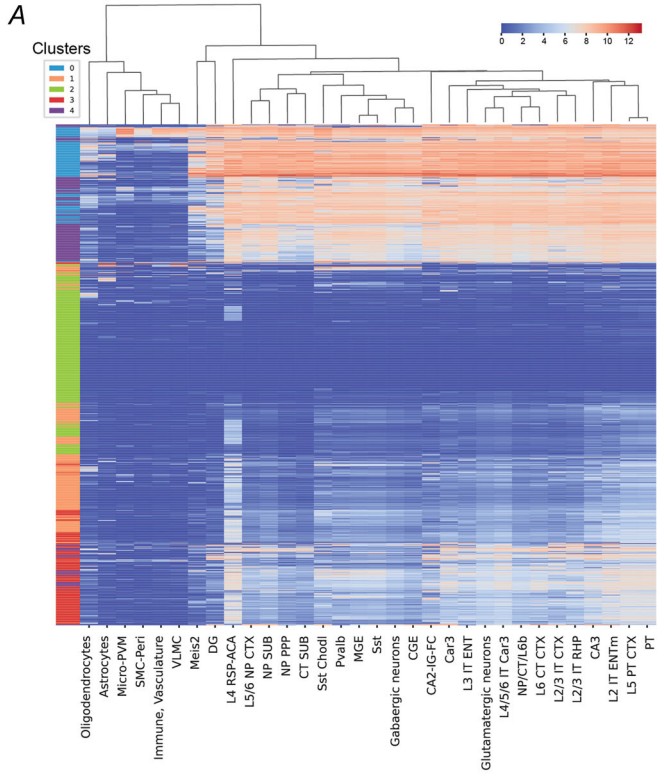

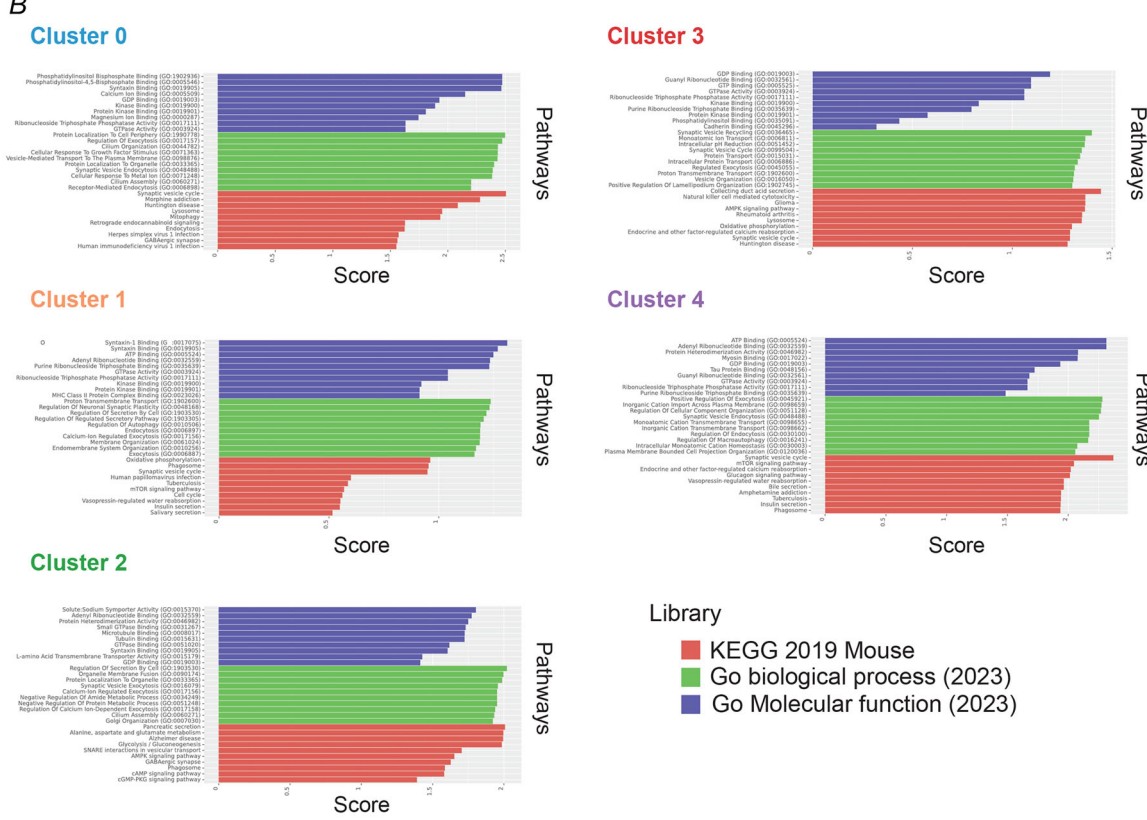

**Figure 1. Expression of genes encoding for synaptic vesicle proteins in different cell types in the mouse cortex and hippocampus, highlighting the possibility that the synaptic vesicle cluster proteome exhibits molecular diversity between neuronal subtypes**

*A*, in this heatmap, the rows represent genes corresponding to synaptic vesicle proteins identified in a synaptic vesicle fraction of the mouse brain by mass spectrometric analysis. Columns represent hierarchical cell types

annotated in the single-cell Mouse Whole Cortex and Hippocampus 10x dataset from the Allen Brain Map (https://portal.brain-map.org). The cells contain the averaged expression levels of these genes in the Mouse Whole Cortex and Hippocampus 10x dataset. The heatmap displayed clear clustering patterns among cell types and genes. Genes were clustered based on their average expressions in the cell types using k-means clustering into five groups. The left colour column shows the clustering results as a coloured column on the left side of the heatmap (clusters 0–4, blue, orange, green, red and purple). *B*, the top 10 pathways for gene sets in each of the five identified clusters of synaptic proteins. Enrichment analysis was conducted using gene set enrichment analysis and the standard databases 'KEGG_2019_Mouse', 'GO_Molecular_Function_2023' and 'GO_Biological_Process_2023', with the average abundance of the genes in the synaptic protein list serving as the score (Hendriks et al., 2018).

**Table 1. Established and theorized involvements of the SVC in regulation of synaptic biology**

| | Known | Hypothesized |
|---|---|---|
| **Direct SVC involvement** | • Cell adhesion molecules are delivered on SV precursors<br>• Vesicle lipid composition affects the morphology of LLPS clusters<br>• SVs bind AZ proteins<br>• Proto-SVs recruit exo- and endocytosis cofactors<br>• SVC acts as a buffer for various soluble proteins, and regulates their mobility<br>• SVs and synapsin recruit actin and affect its polymerization dynamics<br>• SVC proteins interact with tubulin and affect microtubule dynamics<br>• SVC regulates the sizes of vesicle pools<br>• SVC contains numerous regulators of ER dynamics | • SV precursors are a source of membrane for axon growth<br>• Proto-SVs ensure proper relative positioning of SVC and AZ<br>• SVC recruits enzymes involved in post-translational modifications<br>• SVC directs nanoscale organization of the AZ<br>• SVC controls the balance between exo- and endocytosis<br>• SVs influence exo- and endocytosis by controlling the local availability of specific cofactors<br>• SVs regulate mitochondria positioning by competing with them for transport motors or by modulation of their release from microtubules<br>• SVC affects the ER via direct contact sites |
| **Indirect SVC involvement** | • Release of agrin initiates postsynaptic receptor clustering<br>• Neurexin–neuroligin interactions contribute to PSD95 recruitment and PSD formation<br>• PSD formation leads to actin rearrangements and morphological maturation of the spine<br>• SVC contains various ECM binding partners | • Accumulation of recycling cofactors allows maturation of proto-SVs into SVs<br>• Changes in SVC size affect postsynaptic cytoskeleton<br>• SVC regulates ER distribution by affecting the local cytoskeleton<br>• SVC affects spine apparatus via its effects on the PSD and postsynaptic cytoskeleton<br>• SVC controls synaptic stability via its effects on cytoskeleton and/or cell adhesion molecules, and by the modulation of BDNF release |

abundance of synaptic-like vesicles are produced, with limited quality control, ending up with various organelles that contain different SV markers (Bonanomi et al., 2006; Nozumi & Igarashi, 2018), or, as Rab3, on organelles that mostly transport AZ components (Zhai et al., 2001). For simplicity, we will term these organelles proto-SVs.

The proto-SVs are certain to also contain adhesion molecules involved in synaptic contacts and implicated in synaptic function. Such molecules have long been known to interact with SVs, such as CHL1 (Leshchyns'ka et al., 2006), whereas others are highly abundant in SVs, such neural cell adhesion molecule (NCAM), which is found in the top 100 most abundant SV proteins (Taoufiq et al., 2020) and in the top 20 molecules in the axonal growth cones (Poulopoulos et al., 2019), and which participates in both SV recycling and synapse formation (Polo-Parada et al., 2001). Other important adhesion molecules found in SVs include the whole synaptic cell adhesion molecule set (1–4) (Zhang & Wei, 2024), as well as neurexins, reticulons, neuronal pentraxin 1, protein tyrosine phosphatase receptor sigma and protein tyrosine phosphatase receptor type D (Südhof, 2021). If the mature

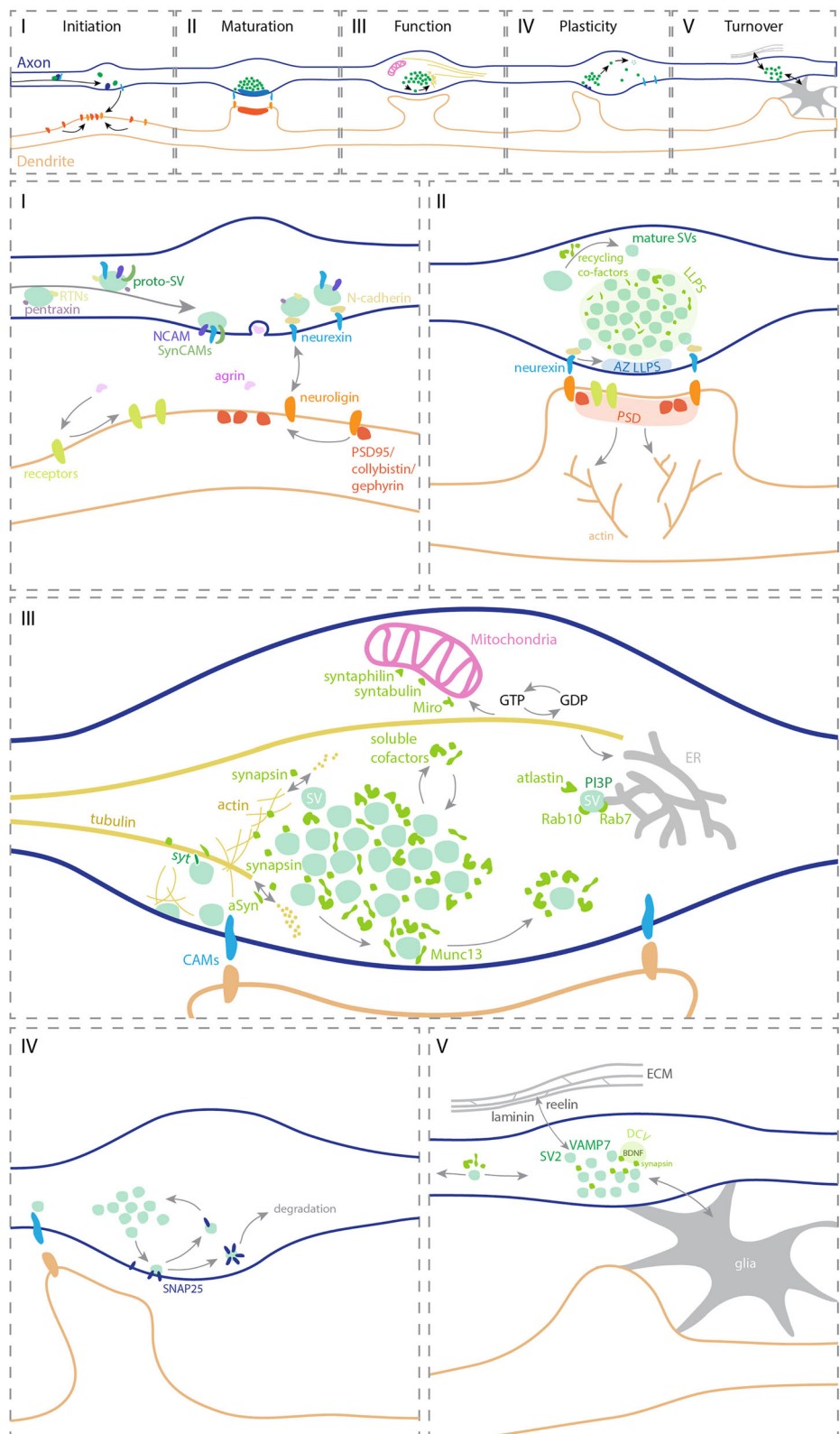

**Figure 2. Schematic representation of processes and mechanisms by which the SVC regulates various aspects of synaptic physiology**

Top: overview of different steps in synaptic function, from contact initialization to synaptic turnover. The zoomed-in views indicate the following: (I) Synapse initialization, for which the flow of proto-SVs is essential because it brings the different adhesion proteins and scaffolds to the location of the future synapse. (II) During synapse maturation,

SVC dynamics regulate the size and the initial organization of the synapse. (III) The SVC controls synaptic function, by co-ordinating the location and activity of different organelles, including the ER and mitochondria, and by regulating processes as exo- and endocytosis and cytoskeletal dynamics. (IV) Synapse plasticity is influenced by the SVC, through the removal and/or addition of elements such as SVs and adhesion proteins. (V) The stabilization of a synapse, or its removal, depends on the SVC interaction with brain components such as the ECM or glia.

SVs contain such molecules, it is more than probable that the proto-SVs used in axonal growth also contain them, and add them continually to the axon surface, through fusion to the axonal plasma membrane.

A typical example are the neurexins, one of the largest families of synaptic adhesion proteins. According to our hypothesis, the flow of proto-SVs, containing SV markers along with adhesion proteins, serves to place neurexins all along the growing axon. When they encounter post-synaptic partners, their diffusion in membranes would be impeded, enabling them to concentrate locally other presynaptic partner proteins, from scaffolds to channels (Luo et al., 2020), which in turn would recruit further neurexins. When a patch of neurexin molecules thus concentrates at the junction with a postsynaptic partner, they would then recruit further proto-SVs on the pre-synaptic side (Dean et al., 2003). These vesicles would bring along more neurexins, thereby leading to synapse growth. The accumulation of neurexin proteins on the presynaptic side is then expected to trigger the clustering of their postsynaptic binding partners, such as neuro-ligins, thus strongly contributing to synapse formation. In turn, neuroligins interact with postsynaptic scaffolds, including postsynaptic density protein 95 (PSD95) (neuroligin 1, at excitatory synapses) or collybistin and gephyrin (neuroligin 2, at inhibitory, dopaminergic and cholinergic synapses) (Südhof, 2017), presumably leading to their accumulation in the postsynapse. This type of interaction chain (neurexin–neuroligin–scaffolds) has been well documented for PSD95, for which recruitment depends on neuroligin 1 clustering and neurexin 1-beta-neuroligin 1 interactions (Giannone et al., 2013).

According to this view, the massive flow of proto-SVs in the growing axon has a major influence on the initial synapse formation. An apparent flaw of this reasoning is that it omits the initial contribution of the postsynapse to the entire process. Neuroligin has been shown to initiate the clustering of neurexins (Dean et al., 2003), but what makes the postsynapse have the adhesion molecules in the place where they are needed? Although this might differ between cell and synapse types, some lines of evidence indicate that this might also be regulated by the pre-synaptic vesicles (albeit not by neurotransmitter release, since normal postsynapses are found in the absence of pre-synaptic release; (Sando et al., 2017; Sigler et al., 2017)). In mammalian neuromuscular junctions, the clustering of ACh receptors, which are uniformly distributed along the postsynaptic (muscle cell) membrane prior to synapse formation (a pattern also followed by adhesion molecules) (Barrow et al., 2009) is induced by the release of agrin from axonal vesicles (Reist et al., 1992). Similarly, axonal agrin has been shown to be important for synaptogenesis in neuronal cultures (Gingras et al., 2002). Thus, agrin can be seen as a presynapse-derived inducer of synapse formation.

Overall, we conclude that the initial flow of proto-SVs in the axon has a major influence on the formation of the synapse, with the vesicles blindly carrying many of the necessary components for synapse initiation. The proto-SVs are not sufficient for generating mature synapses because they rely on scaffold components to form the SVC at the synaptic site (Marcó de la Cruz et al., 2024). Nonetheless, synapse initiation would be impossible in the absence of the flow of proto-SVs.

## Synapse maturation

Once the contact between the immature pre- and post-synapse is established by the adhesion molecules, a cascade of intracellular interactions ensues, leading to the formation of the typical synaptic structures, such as the AZ, SVC and the PSD. In recent years, experimental evidence accumulate that the assembly of all three of these components may be mediated by LLPS (Chen et al., 2020; Sansevrino et al., 2023). Evidence for LLPS mechanism is provided by minimal in *in vitro* systems, showing that SVs and synapsin is sufficient for condensate formation (Milovanovic et al. 2018), with the morphology of condensates and the intervesicle distance regulated by the lipid-to-protein ratio (Alfken et al., 2024). Furthermore, micropipette aspiration showed that elastic forces can be present in the vesicle-synapsin condensates, which may also account for limits in the condensate growth and size. As already described above, the presynaptic adhesion molecules, brought by the SV precursors, can interact with and recruit the SVs, which then further bring other proteins, such as synapsin, alpha-synuclein (aSyn), and multiple exo- and endocytosis cofactors. Although the clustering of the SVs relies mostly on the synapsin-driven LLPS (Hoffmann et al., 2023), the correct mutual positioning of the forming SVC and the future AZ is probably ensured by early interactions between the SVs, AZ scaffolds and presynaptic adhesion proteins. Both neurexins and SVs contribute to the eventual assembly of the AZ; for example, by organizing the calcium channels and recruiting various AZ proteins (Luo et al., 2020; Miki et al., 2017a; Regus-Leidig et al., 2009), which then

undergo LLPS to form the mature AZ (Emperador-Melero et al., 2021; McDonald et al., 2020, 2023). Even large scaffolds such as bassoon and piccolo are relatively abundant on SVs, implying a substantial role of these organelles in the potential recruitment of AZ components to the synaptic site, or in their local maintenance (Taoufiq et al., 2020). The scaffolds, in turn, appear to be strongly involved in recruiting further adhesion proteins (Frankel et al., 2023). Similarly, the neuroligins, now clustered on the postsynaptic side, recruit PSD95 and postsynaptic receptors, encouraging the LLPS-driven formation of the PSD (Bai & Zhang, 2022; Hosokawa & Liu, 2021; Poulopoulos et al., 2009; Soykan et al., 2014).

On the presynaptic side, the accumulation of vesicles leads to the simultaneous recruitment of numerous exo- and endocytosis cofactors, for which the presence at physiologically accurate copy numbers allows the proto-SVs to turn into proper SVs, by recycling. Although the precursor vesicles can also be released and consequently are recycled along the axon, this recycling is unspecific and results in the formation of immature vesicles that do not yet have the stereotypic composition of the SVs. By contrast, endosomal sorting in the presynaptic bouton allows for control of SV morphology and composition (Hoopmann et al., 2010; West et al., 1997). This sorting must be highly specific, to ensure, among other features, the inclusion of the correct SV protein copy numbers (Imig et al., 2014). This is especially evident for key SV proteins such as the vATPase, vGlut1, synaptotagmin and SV protein 2 (SV2), with the latter molecule being the most extreme case, with 97% of SVs bearing exactly five molecules of SV2 each (Mutch et al., 2011). Such specificity can be achieved as a result of the SVC concentrating the cofactors needed for the recycling, including various adaptor proteins (e.g. AP1, AP2, CALM, AP180, stonin), small GTPases (e.g. Rab5, Arf6), clathrin, nexins and others (Kim & Ryan, 2009; Kononenko et al., 2013; Mahoney et al., 2006; Mignogna & D'Adamo, 2017; Nonet et al., 1999; Sheehan & Waites, 2019; Sun et al., 2020).

The activity of the recycling cofactors is subject to regulation through, in large part by post-translational modifications (Ilic et al., 2022; Kawabe & Brose, 2011), from which phosphorylation events at particular sites on the synaptic proteins are the most studied (Engholm-Keller et al., 2019; Silbern et al., 2021). Similar to many other soluble proteins, many enzymes required for such modifications must be delivered to the presynaptic termini on precursor vesicles, and may then also be recruited into the SVC. The co-ordinated activation of these modifying enzymes is probably orchestrated by the local synaptic environment, specifically by the activity history, energy status and network inputs of the corresponding synapses. The activity-dependent messengers probably involved include $Ca^{2+}$, ATP and other 'energy-status indicators' (metabolites), second messengers (e.g. cAMP), or signalling lipids (e.g. diacyl glycerol, phosphoinositides). Major synaptic post-translational modifying enzymes are kinases and phosphatases, many of which are regulated by the activity-dependent messengers listed above, such as protein kinase C, calcium/calmodulin-dependent protein kinases, protein kinase A, protein phosphatase 1 or 2, and in turn affect SVC dynamics by controlling the phosphorylation status of major pre-synaptic proteins as synapsin1 or aSyn (Sansevrino et al., 2023). Similar scenarios arise with other post-translational modifications, such as ubiquitination (Pinto et al., 2016).

Interestingly, no single protein has been identified for which knockout would completely abolish the formation of an AZ, suggesting that a co-operative and co-ordinated action of the factors mentioned above is required (Acuna et al., 2016). The removal of multiple proteins (Marcó de la Cruz et al., 2024) will drastically perturb synapse formation, again suggesting that the interplay of several molecules is always necessary. How is the timely delivery of sufficient quantities of these components to the synapse achieved, without a master regulator protein? Considering all of the key proteins needed for AZ assembly (RIMs, neurexins, leukocyte common-antigen related receptor tyrosine phosphatases, liprins) have been reported to interact with the SVs (Fernández-Busnadiego et al., 2013; Wilhelm et al., 2014), it is reasonable to suggest that the SVs (and their precursors) act as important regulators of this process, at least by bringing molecules to the AZ sites (Rizalar et al., 2023).

The SVC probably also plays an important role in the correct nanoscale assembly of the AZ (as opposed to simply participating in the recruitment of such molecules in the synapse). A prominent feature of the AZ is the presence of protein clusters containing RIM, Munc13 and $Ca^{2+}$ channels (Böhme et al., 2016). RIM and RIM-BP are suggested to drive the AZ formation by forming condensates via LLPS (McDonald et al., 2020; Wu et al., 2019), which then recruit $Ca^{2+}$ channels (Grauel et al., 2016; Jung et al., 2015; Kaeser et al., 2011) and interact with SVs (Wu et al., 2021). RIM binds SVs (or Rab3 on the SV surface) and tethers them to the membrane (Fernández-Busnadiego et al., 2010), implying that its distribution is interdependent with that of the vesicles. In turn, RIM determines the organization of Munc13 in functional clusters (Sakamoto et al., 2018). Therefore, although the AZ organization probably relies on its own LLPS processes and interactions with various molecules (e.g. adhesion proteins), the SVC interacts with most of these proteins, thereby directing their positioning.

A very important question is whether the organization of $Ca^{2+}$ channels is dependent on the SVC. Several possible topographies have been suggested, including random positioning of $Ca^{2+}$ channels and SVs

(Meinrenken et al., 2002), $Ca^{2+}$ channels forming a ring around SVs (Wang et al., 2009), positioning of SVs at the perimeter of $Ca^{2+}$ clusters (Chen et al., 2024; Nakamura et al., 2015; Wong et al., 2014) or in a one-to-one stoichiometry (Miki et al., 2017b), or an exclusion zone model where $Ca^{2+}$ channels do not cluster but are simply excluded from a 50 nm zone around docked SVs (Rebola et al., 2019). These different scenarios probably reflect different implementations of relative topography of $Ca^{2+}$ channels and release sites at different synapses (Eggermann et al., 2011) and can coexist in the same circuitry (Rebola et al., 2019) or even in the same cell (Özçete & Moser, 2021). Nonetheless, other than the fact that SVs need to be coupled to $Ca^{2+}$ channels, functionally, and the numerous observations of interactions between SVs, RIM, RIM-BP and channels, no clear answer is available on the SV influence the channel distribution. Advances in electron microscopy and in optical nanoscopy are now enabling the direct mapping of the nanotopography of $Ca^{2+}$ channels and release sites, at increasing levels of resolution (Böhme et al., 2016; Grabner et al., 2022; Sakamoto et al., 2018), suggesting that answers to such questions may be forthcoming in the near future.

On the postsynaptic side, the sequence of processes during synapse formation are less conclusive, but it is very probable that the recruitment of scaffold proteins, as discussed in the previous section, sets into motion mechanisms that culminate with the formation of functional receptors. The PSD is a densely packed ensemble of many different proteins (Bayés et al., 2011; Li et al., 2017; Roy et al., 2018; Sheng & Hoogenraad, 2007), probably formed via LLPS (Chen et al., 2020). The scaffold-based LLPS phase organizes receptor clustering and potential subdomains in the PSD (Hosokawa & Liu, 2021). A minimal PSD was reconstituted *in vitro* by condensation of four PSD proteins, PSD95, GKAP, Shank and Homer3a, which were truncated to primarily contain the multivalent protein interaction motifs (Zeng et al., 2018). This minimal PSD, when mixed with the GluN2B PDZ binding motif fused to a coiled-coil domain to form tetramers, clustered these NMDA receptor proxies, mimicking clusters of receptors as observed in synapses (Goncalves et al., 2020). These experiments revealed a self-organizing potential of PSD proteins that may underlie the nano-organization of the synapse. Alternating protein compositions further induced subdivisions of these protein complexes. The long-term potentiation (LTP)-associated kinase CaMKIIa formed after $Ca^{2+}$ activation condensates with the GluN2B C-terminus (Hosokawa et al., 2021). When mixed with PSD95 and the C-terminus of the AMPA receptor (AMPAR) auxiliary subunit stargazin, the condensates separated, with PSD95 preferentially binding to stargazin and not the GluN2B construct. Thus, $Ca^{2+}$ and/or kinase activity might change the binding preference of PSD95 with different receptor clusters, revealing some of the dynamics that might occur during synaptic plasticity.

These results are consistent with the role of PSD95 as a signalling scaffold in which the receptor recruits it for linking signalling pathways (Liu et al., 2017; Xu et al., 2008). PSD95 promotes AMPAR incorporation into nascent synapses but not mature synapses (Béïque et al., 2006; Huang et al., 2015; Levy & Nicoll, 2017; Schnell et al., 2002). Furthermore, this function is inhibited by the paralogous protein PSD93, indicating their pivotal role in AMPAR regulation during synapse maturation (Favaro et al., 2018). Notably, AMPARs appear before PSD95 in the synapse, indicating that, although PSD condensates can cluster receptors, the initial trigger for protein recruitment might originate from the membrane receptors or adhesion proteins (Subramanian et al., 2019). This scenario would allow the trigger for postsynaptic differentiation to originate from the presynase, and would also enable the latter to modulate postsynaptic size.

This accumulation of scaffolding proteins and receptors will in turn influence the activity of the presynaptic site (Tracy et al., 2011) and will recruit actin in the postsynapse, yielding significant morphological changes during the progression from thin filopodia to classical mushroom-type dendritic spines, which relies on actin cytoskeleton rearrangements. These rearrangements can be induced by the PSD via the interaction of PSD95 with $\alpha$-actinin (Matt et al., 2018), by the WAVE complex, a group of proteins that interact with the branching protein complex ARP2/3 and actin to nucleate new filaments, which is located at the PSD (Chazeau et al., 2014), or even by the PSD condensate itself, without relying on any regulatory proteins (Chen et al., 2023). Importantly, the formation of the synapse (e.g. through the neuroligin/neurexin coupling) will constrain the membrane around the synaptic cleft, which in turn shapes the membrane of the whole spine and the counterforces it puts up against deformation. These counterforces can in turn control actin polymerization and elongation via the Brownian ratchet mechanism (Mogilner & Oster, 1996). Overall, this leads to a self-organization of the actin-membrane geometry system and may bring actin to a critical state (Bonilla-Quintana et al., 2020, 2021), which enables the spine to react faster to stimulation (Bonilla-Quintana et al., 2021).

Finally, as both the pre- and postsynapse arrange their key components, presynaptic activity will turn this synapse into a unit capable of transferring information, the regulation of which is explained in the following sections.

## Structural and functional aspects

It is assumed that the SVC directly controls synaptic activity, via its modulation of various soluble proteins (Rizzoli, 2014). As the SVC grows, it keeps recruiting

and accumulating the various soluble cofactors needed for vesicle recycling. These cofactors engage in a multitude of different types of functional complexes involved in various aspects of synaptic function (Koopmans et al., 2019; Sorokina et al., 2021; Xu et al., 2021), and their effective retainment in the synaptic bouton is crucial for the maintenance of these processes. The binding of these proteins to the SVs not only enriches the cofactors in the pre-synapse, maintaining adequate protein abundances, but also regulates their mobility levels, thereby regulating their availability for recycling and controlling the neuro-transmitter release rates (Reshetniak & Rizzoli, 2021; Reshetniak et al., 2020). At the same time, the copy numbers of cofactors, although proportional to the number of vesicles, do not linearly correlate with the size of the SVC. As the SVC size increases, the amount of cofactors contained tends to saturate, so that larger synapses contain fewer cofactors per SV (Wilhelm et al., 2014). This appears to be a passive adaptation that is especially relevant for the endocytosis cofactors, for which copy numbers are far more limiting than those of the exocytosis cofactors, implying that larger synapses should still exocytose efficiently, but would not be efficient during endocytosis, leading to the diffusion of vesicle components from their AZs. Ultimately, this mechanism should, in principle, lead to a reduction in synapse size, prohibiting synapses from becoming unnaturally large. These observations imply that the SVC, via its modulation of various exo- and endocytic proteins and scaffolds, provides a synapse-specific link between synaptic exo- and endocytosis, for each individual synapse. Thus, the SVC appears as an important regulator of not only synapse size, but also of synaptic activity because it may effectively control the balance between exo- and endocytosis at given activity levels. Should the activity levels surpass the end-ocytosis capacity at the respective synapse (controlled by the SVC via endocytic cofactor copy numbers), activity is slowed down considerably; for example, via problems with release site clearance (Haucke et al., 2011).

In addition to exo- and endocytosis cofactors, cyto-skeletal components, such as actin, are also highly enriched in the SVC (Reshetniak et al., 2020; Wilhelm et al., 2014). Actin has been shown to form distinct functional arrangements within the bouton, and is assumed to contribute to the maintenance of the SVC, by limiting the mobility of the SVs, at the same time as having an important role in SV endocytosis (Bloom et al., 2003; Rizzoli & Betz, 2005; Sankaranarayanan et al., 2003). Modern views of actin assembly in boutons (Bingham et al., 2023) refer to multifilament corrals, surrounding the SVC, filamentous rails within the SVC, presumably used as guides during vesicle transport for exocytosis, and actin meshes at the presynaptic membrane, which are probably employed in endocytosis (Oevel et al., 2024; Soykan et al., 2017). It is not clear how the various actin

morphologies are achieved in the synaptic bouton, but a number of observations indicate a role of the SVC in this process. First, actin can be recruited to the presynapse by synapsin: the major soluble component of the SVC (Bloom et al., 2003). Second, actin localizes to the patches of SV proteins on the plasma membrane, indicating that other SV proteins also affect actin distribution outside the SVC (Dason et al., 2014). Lastly, actin polymerization kinetics can be modulated by SVs and synapsin *in vitro* (Benfenati et al., 1992; Ceccaldi et al., 1995; Chieregatti et al., 1996), suggesting a direct link between actin filament assembly dynamics and the SVC.

Although less information is available of the pre-synaptic tubulin, close association between the SVC and tubulin is also expected. Mainly seen as 'tracks' for active transport, microtubules are also known to penetrate the SVC and associate with the SVs (Babu et al., 2020; Grey & Katz, 1997; Grey et al., 1982). Tubulin directly inter-acts with several SVC proteins, including synaptotagmin, synapsin and aSyn, with the latter two being able to directly affect microtubule dynamics (Baines & Bennett, 1986; Cartelli et al., 2016; Honda et al., 2002), suggesting that the SVC might also organize presynaptic tubulin cytoskeleton. Lastly, dynamic microtubules are nucleated in synaptic boutons, in an activity-dependent fashion, strongly indicating a link to the SVC or SVC-dependent activity.

Finally, even less information is available for pre-synaptic intermediate filaments. However, it is worth pointing out that many septin isoforms are found on purified SVs (Taoufiq et al., 2020), in more than trace levels. Because septins are involved in multiple steps in SV recycling, such as priming and exocytosis (Dolat et al., 2014), their abundance in the SVC probably regulates functional interactions.

In addition to direct effects on the cytoskeleton on the presynaptic side, the SVC can also indirectly affect it in the postsynapse because the postsynaptic actin cytoskeleton undergoes significant morphological changes in response to PSD formation. Because the size and position of the PSD are directly influenced by the SVC, the subsequent cytoskeletal rearrangements can be seen as an indirectly regulated by the SVC (as already indicated in the previous section).

In functional terms, the SVC will influence, for example, the endocytosis mechanisms. As SVs approach the plasma membrane prior to their release, they bring along the SV-bound cofactors. Even in conditions when these proteins are not used for recycling of this specific vesicle, such as during transient docking without fusion (Kusick et al., 2020), they would affect their immediate surroundings. The local conditions induced by the cofactors and cytoskeleton molecules organized by the vesicles can then determine the various processes taking place during or after fusion, such as different modes of

neurotransmitter release or endocytosis (Bolz et al., 2023; Chanaday et al., 2019; Watanabe & Boucrot, 2017; Wu & Chan, 2022).

## Vesicle pool considerations

The SVC is also able to regulate the different vesicle pools, the abundance and behaviours of which are controllers of short-term plasticity. Co-ordinated inter-actions between SV proteins, AZ scaffold molecules and components of the presynaptic release machinery control the nanoscale organization of synaptic vesicles at AZs, as well as their functional maturation to a fusion-competent ('primed') and readily-releasable state (Varoqueaux et al., 2002). Prior to stimulus-evoked soluble *N*-ethylmaleimide-sensitive factor attachment protein receptor (SNARE)-mediated fusion at AZ release sites, vesicles transition through multiple, reversible molecular states (He et al., 2017; Imig et al., 2014; Kusick et al., 2020; Papantoniou et al., 2023) at rates sensitive to presynaptic calcium levels and second messengers. Importantly, this complexity supports a range of mechanisms via which SVC organization and the efficacy of presynaptic neurotransmitter release can be dynamically/plastically and purposefully fine-tuned to alter synaptic strength.

Correspondingly, numerous co-determinants of synaptic short-term plasticity characteristics directly related to the dynamic organization of the SVC have been postulated, including (1) the availability of primed vesicles within the readily-releasable pool (Imig et al., 2020; Maus et al., 2020) and their rates of replenishment during sustained synaptic activity (Lipstein et al., 2013); (2) the ratio of tightly docked (TS) to loosely docked (LS) vesicle states (Neher, 2024; Neher & Brose, 2018); (3) the occupancy of AZ release sites (Malagon et al., 2020; Pulido & Marty, 2017); (4) the species of vesicular $Ca^{2+}$ sensor (Geppert et al., 1994; Jackman et al., 2016) and coupling distances to voltage-gated calcium channels (Rebola et al., 2019; Vyleta & Jonas, 2014); and (5) the efficiency with which AZ docking sites are molecularly restored following synaptic vesicle fusion (release site clearance) (Rajappa et al., 2016).

Neuronally expressed priming proteins of the Munc13 family, Munc13-1, bMunc13-2, ubMunc13-2 and Munc13-3 (Brose et al., 2000), are essential for the formation and replenishment of the readily-releasable pool of synaptic vesicles (Imig et al., 2014; Varoqueaux et al., 2002) and are also buffered by the SVC (Reshetniak et al., 2020). Different priming models have evolved to interpret the strength and time-course of synaptic trans-mission measured by electrophysiological approaches in the context of synaptic ultrastructure and the spatial distribution of SVs resolved by electron microscopy. In the two-step priming model, vesicle priming occurs in dynamic equilibrium at AZ release sites via reversible transitions through two states; empty release sites are initially occupied by vesicles in a LS state that can subsequently mature to a TS and fusion-competent state (Neher, 2024; Neher & Brose, 2018). The TS state is considered analogous to the Munc13- and SNARE-dependent docked vesicle state as visualized by electron tomography (ET) (Imig et al., 2014). Release probability is considered to be homogenous at a fixed number of release sites and forward priming reaction rates are promoted by activity and elevations in pre-synaptic $Ca^{2+}$ concentration. According to this model, the plasticity characteristics of depressing ('phasic') and facilitating ('tonic') synapses can be accounted for by high and low ratios of TS to LS vesicles in resting synapses, respectively. Although the notion that synaptic strength and plasticity characteristics manifest as differences in SV distribution at AZs has been contentious (Xu-Friedman et al., 2001), a comparative ultrastructural analysis revealed that hippocampal Schaffer collateral ('phasic') and mossy fibre ('tonic') synapses were characterized by high and low ratios of docked to 'tethered' vesicles, respectively (Maus et al., 2020). This latter finding appears to provide structural support for the two-step priming model; however, it remains to be clarified whether (1) TS and LS states can be reliably discriminated at the ultrastructural level and (2) whether synapse-specific differences in numbers of docked vesicles reflect differences in the availability of release sites or rather differences in the occupancy of release sites. Nevertheless, current dogma, based on widely accepted depletion models, attribute short-term depression to an exhaustion of docked and molecularly primed SVs (Zucker & Regehr, 2002). Direct ultrastructural evidence in support of this has been demonstrated in hippocampal mossy fibres using time-resolved electron microscopy to correlate steady-state synaptic depression with depletion of docked/primed and membrane-proximal vesicle pools (Imig et al., 2020).

An alternative model postulates that parallel priming pathways generate fusion-competent vesicles with heterogenous release probabilities (Eshra et al., 2021). Heterogeneity within the readily releasable pool may have spatial (e.g. $Ca^{2+}$-channel coupling) (Rebola et al., 2019; Vyleta & Jonas, 2014) and/or molecular (i.e. vesicular $Ca^{2+}$-sensor) (Jackman et al., 2016) origins, with iso-forms of the synaptotagmin family being of particular interest, given differences in their $Ca^{2+}$-binding and -unbinding kinetics (Hui et al., 2005) and their postulated roles in presynaptic transmitter release.

Overall, all models of priming would be influenced strongly by the SVC, which regulates the spatial organization of the different components (SVs, $Ca^{2+}$ channels), as well as the nature of the different SVs. Similarly, the SVC is self-sufficient in generating a reserve

pool, by maintaining a system where vesicles composed of older molecules can be detected and have a lower probability of being released, leaving younger vesicles to constitute a recycling pool (Truckenbrodt et al., 2018).

All of these effects contribute to having the SVC as a regulator of how much, and how quickly, neuro-transmitter can be released by a synapse, which is an obvious controller of the postsynaptic response, as well, and hence determines the contribution of the synapses to information transfer.

## The case of synaptic mitochondria

As for all the other cellular compartments, the presynaptic bouton, as well as the postsynaptic spine, requires energy to maintain its activity. Processes such as translation, active transport or protein degradation, which occur both at the synapse and in dendrites and axons, have been shown to be energy intensive (Hirokawa & Takemura, 2005; Hua et al., 1997; Moldave, 1985). The ATP synthesis thus needs to be local and tuned to activity, as supported by the dendritic mitochondria profile (López-Doménech et al., 2016) and synaptically-localized mitochondria (Seager et al., 2020). However, ATP in the presynapse is probably not just an energy source for vesicle acidification or other active transport processes. ATP has been shown to promote or impede LLPS-induced protein condensate formation in a concentration-dependent, but energy-independent manner (Dang et al., 2021; Hayes et al., 2018; Kang et al., 2018; Patel et al., 2017; Ren et al., 2022; Song, 2021). In addition, it is required for protein phosphorylation, with one prominent target being synapsin, for which the phosphorylation state affects its SV-clustering ability and, as a result, vesicle mobility (Chi et al., 2003). Another target is aSyn because it was recently shown that its phosphorylation on S129 can act as a physiological regulator of neuronal activity (Ramalingam et al., 2023). Thus, ATP plays an important role in the organization of the SVC, affecting vesicle pool behaviour, in addition to its role as an energy source. Improper ATP concentrations might cause a dispersion of the SVC, or such a drastic stabilization that no effective vesicle recycling might take place, with both scenarios resulting in synaptic failure. Therefore, presynaptic ATP production needs to be well balanced, and consequently the respiratory activity and presence of mitochondria in the presynapse is regulated.

In hippocampal neurons, ~35% of presynapses are occupied by mitochondria, whereas, in the auditory system, up to 80 % of presynapses were shown to contain mitochondria, possibly reflecting the distinct energy demands of the presynapses (Hintze et al., 2024; Smith et al., 2016). The subset of mitochondria that remains stationary at presynapses has been proposed to support prolonged synaptic activity (Smith et al., 2016).

These resident synaptic mitochondria are more sensitive to calcium fluxes and thus mitochondrial ATP production might be regulated by short-term changes in presynaptic energy demand (Datta & Jaiswal, 2021; Patron et al., 2019). In addition, long-term synaptic activity has been shown to impact the mass and inner membrane architecture of synaptic mitochondria (Hintze et al., 2024; Thomas et al., 2019).

Mitochondria are presumably actively recruited to presynaptic sites through a short and transient increase in energy demand. Inhibition of presynaptic activity enhances mitochondria mobility along axons, whereas synaptic overactivity increases the tendency of mitochondria to localize at presynaptic sites (Chang et al., 2006; Chen & Sheng, 2013; Sheng, 2014). How could such activity-dependent changes in mitochondrial localization be mediated?

The mitochondria-associated protein Miro establishes interactions of the organelle with microtubules (Li et al., 2024). This interaction is crucial for the transport of mitochondria in the axons. Spikes of calcium concentrations at presynapses induce conformational changes of Miro on en-passant mitochondria. This conformational change interferes with Miro-microtubule binding and further promotes interactions of the protein syntaphilin with mitochondria, which could ultimately lead to mitochondrial stalling at sites with higher synaptic activity (Kang et al., 2008; Silva et al., 2021).

Another possible factor could be a competition of mitochondria and the SVs for motor proteins. SV precursors are mainly transported by specific motor proteins (Okada et al., 1995), whereas mitochondria are trafficked by a set of multiple motors, which are partly overlapping with those used by the SVs (Hirokawa et al., 2010). Hence, we hypothesize that the competition for the motors might influence the localization of the mitochondria at presynaptic sites. A few candidate proteins that act at the interface between mitochondria and axonal vesicle transport have been identified, including syntabulin and syntaphilin (Cai et al., 2005; Kang et al., 2008), although these proteins interact with dense core vesicles and plasma membrane protein syntaxin, rather than the SVs directly.

Long-term positioning of mitochondria in the presynapses might be regulated by modulation of their release from the microtubules. Several mechanisms influencing the un-linking of transport cargoes have been observed, many relying on the changes in motor affinity to the cargo or microtubules (Hirokawa et al., 2010). Such changes could be initiated by posttranslational modifications of the motors themselves, or by modifications to the cargoes. For example, release of Rab3 from a kinesin can be triggered by its GTPase activity because it is transported preferentially in a GTP- rather than GDP-bound state (Niwa et al., 2008). The local availability of Rab-GAPs, Rab-GEF, kinases or phosphatases would therefore affect the active

transport or release of various cargoes. By accumulating a rich selection of such proteins and controlling their concentration and mobility, the SVC would create specific conditions that favour the unlinking of required organelles in a given bouton (albeit not their introduction within the SVC because they appear to be largely excluded from the SV-based phase, in many synapse types (Denker et al., 2011a; Wilhelm et al., 2014)).

Another important aspect of mitochondria biology is their peculiar protein biogenesis. The mitochondrial proteome comprises ∼1500 different proteins. Although most proteins are nuclear-encoded and imported into mitochondria, in metazoans, 13 proteins are encoded by mitochondrial DNA. The composition of the mitochondrial proteome can differ significantly between cell types, aiming to cope with cell type-specific metabolic demands (Sung et al., 2020). Little is known about the kinetics by which the mitochondrial proteome is replenished, especially in relation to the complex organization of the neuron. In this context, it has been shown that mitochondrial proteins are translated at the synapse and are imported into mitochondria, demonstrating that proteostatic processes occur spatially separated from the cell body (Kuzniewska et al., 2020). There is evidence for trafficking of mRNAs encoding mitochondrial proteins to the synapse, including on lysosomes, and of their translation on the surface of axonal endosomes (Cioni et al., 2019; De Pace et al., 2024; Gehrke et al., 2015; Harbauer et al., 2022; Liao et al., 2019). In addition, translation of mitochondrial-encoded proteins has been shown to occur in axons, dendrites and at pre- and post-synaptic regions (Yousefi et al., 2021). Based on these observations, it is evident that mitochondrial proteostasis is not spatially limited to the cell body in neurons, but is distributed across the cell. It will be of particular interest, in the future, to describe whether and how this process links to other synaptic functions, and especially to SVC dynamics.

## The endoplasmic reticulum

Another organelle abundantly present in the axons, the endoplasmic reticulum (ER) (Tsukita & Ishikawa, 1976), might have an even more direct relationship with the SVC. First, because of the dependence of the ER distribution on microtubules (Terasaki et al., 1986), the SVC should be able to regulate this distribution by affecting the local cytoskeleton, as described above. Second, the SVC harbors various proteins involved in regulating ER morphogenesis and motility. For example, Rab10, which regulates the dynamics of ER tubules, the predominant ER morphology observed in axons (English & Voeltz, 2013), is one of the most abundant Rab proteins associated with the SVs (Taoufiq et al., 2020). Similarly, atlastins, which are major regulators of ER biogenesis and turnover (Chen et al.,

2019; Rismanchi et al., 2008; Wang et al., 2016), are also found on the SVs. Third, the endoplasmic reticulum is known to form numerous membrane contact sites with various organelles (Wu et al., 2017), including organelles containing Rab7 and phosphatidylinositol 3-phosphate (Raiborg et al., 2015). Both of these molecules are abundant in SVs (Binotti et al., 2021; Lauwers et al., 2016; Wucherpfennig et al., 2003); therefore, the SVs might directly affect the ER via these contact sites. Further contacts to mitochondria may also influence the $Ca^{2+}$ buffering capacity of both organelles, as well as their respective localizations.

On the postsynaptic site, stacks of ER cisterns are organized into the spine apparatus, enriched in the actin-associated protein synaptopodin (Deller et al., 2003). Through its interactions with membrane-associated guanylate kinase with inverted orientation (MAGI) scaffolding proteins (Falahati et al., 2022), synaptopodin provides a link between the spine apparatus and the PSD. Because the assembly of the PSD and the spine actin cytoskeleton are indirectly modulated by the SVC, as we noted above, the morphology and localization of the spine apparatus would, in turn, be also affected by the SVC (Gürth et al., 2020).

Overall, considering the role of the ER in calcium signalling, the ability of the SVC to modulate and organize the synaptic ER might have a substantial role in neuronal physiology, providing an additional avenue for the SVC to regulate synaptic activity.

## Synaptic plasticity

Utilizing the multiple pathways to regulate synaptic activity, the SVC is also clearly involved in synaptic plasticity, with synaptic scaling being the most obvious example. Prolonged periods of enhanced activity lead to frequent use of the SVs. With each round of exo- and endocytosis, these vesicles become progressively more contaminated with plasma membrane-resident SNAP25 molecules, and thus have a lower probability of being released, and are sent for degradation (Jähne et al., 2021; Truckenbrodt et al., 2018). In the absence of compensatory mechanisms and delivery of newly formed vesicles, this ultimately leads to reduction in synapse size, to the extent where the previous activity rates can no longer be maintained. Slowed activity results in the growth of the synapse (although only up to a certain extent, whereas the relative amounts of cofactors are sufficient to recycle the released vesicles efficiently). This leads to a proportional response on the postsynaptic side, by modulating the copy numbers of adhesion molecules, which, in turn, changes the levels of post-synaptic scaffolds. This mechanism would be a key contributor to ensure synaptic reduction or enlargement in response to activity, thus providing a simple and robust

mechanism for homeostatic synaptic plasticity on long time-scales (Turrigiano, 2017; Zierenberg et al., 2018), despite the fact that diverse postsynaptic mechanisms are involved (Turrigiano, 2008). At the same time, activity independent fluctuations in synapse size and protein abundances will also play an important role in synaptic behaviours, presumably assisting in tuning synapse sizes to their functional requirements (Hazan & Ziv, 2020).

On short time-scales, homeostatic synaptic plasticity is complemented by short-term-plasticity (Zucker & Regehr, 2002). It is implemented at each step of the vesicle cycle, with various time-scales. On the very fast timescale (sub-seconds), the depletion of the small readily-releasable pool reduces effectively the synaptic strength. Thereby, excess activity in the network can directly be dampened via an effective reduction of synaptic strength (potentially after a transient facilitation). At intermediate timescales, the limited, but steady, endocytosis rate ensures that high spike rates do not full deplete the vesicles in the recycling pool. Hence, on the scale of seconds and minutes, the architecture of the vesicle cycle regulates activity transmission at the synapse, interestingly in a manner that improves information transfer under natural spike conditions (Rotman et al., 2011).

Classical long-term plasticity [long-term potentiation (LTP) and long-term depression (LTD)] rely, in the majority of central synapses, on highly specific post-synaptic mechanisms such as receptor trafficking or local protein synthesis, which are mediated by various signalling cascades, for which the overall effects are derived from increased or decreased presynaptic activity (Baltaci et al., 2019; Bin Ibrahim et al., 2022; Bliss et al., 2018; Collingridge et al., 2010; Malenka & Bear, 2004). These changes in activity are generally regulated by the SVC, as discussed above, and may rely on multiple structural changes in the presynapse (Monday et al., 2018), implying that the roots of both LTP and LTD can be traced to SVC modulation. Additionally, the role of so-called synapse organizers (cellular adhesion proteins and other components involved in the mechanical stabilization of synapses) in long-term plasticity begins to be recognized (Connor & Siddiqui, 2023). Many of these molecules are brought to the synapse and are organized by the SVC, further highlighting the fundamental role of the SVC in not only initiating synaptic changes during plasticity, but also providing the machinery to execute them.

## Synaptic stabilization and synaptic loss

After their initiation, some synapses may receive stabilization signals, and are maintained, whereas others are lost rapidly. Structural changes associated with long-term changes in synaptic strength have been extensively studied in the past, with a predominant focus on the postsynaptic compartments of small glutamatergic spine synapses (Harris, 2020) motivated by work elucidating postsynaptic loci of long-term plasticity mechanisms in these synapses. However, the emerging view is one in which corresponding changes in presynaptic morphology and SVC organization also contribute to the stability of potentiated synapses (Bourne & Harris, 2011; Chéreau et al., 2017; Rey et al., 2020).

The SVC has a major influence on the stability of synapses, by creating a synapse competition system. Although mature SVs are mainly retained in synaptic boutons in the form of SVCs, it has been long established that individual vesicles are also continuously exchanged between boutons (Darcy et al., 2006; Staras et al., 2010). This effectively creates a situation where multiple synapses compete for the same pool of vesicles and the various cofactors they bind. As individual vesicles are transported to neighbouring synapses and bring additional proteins with them, they contribute to the size and strength of the acceptor synapse, increasing the probability of losing the donor synapses. In agreement with this idea, larger synapses are more stable than small ones, independent of their activity level (Quinn et al., 2019).

The SVC may also regulate many signals implicated in synaptic stabilization. Some rely on the actin cyto-skeleton (Basu & Lamprecht, 2018) or molecules of cellular adhesion (Duncan et al., 2021; Quinn et al., 2017). The interplay between these and the SVs has already been discussed in the previous sections, providing hints toward mechanisms by which the size of the SVC can influence synaptic stability. There are, however, several additional possibilities. For example, brain-derived neuro-trophic factor (BDNF), a crucial molecule with multiple implications in memory, ageing and neurodegenerative diseases (Miranda et al., 2019), is released from pre-synaptic dense-core vesicles in an activity-dependent manner (Dieni et al., 2012; Kohara et al., 2001). The dense-core vesicles not only reside within the SVC (Persoon et al., 2018), but also rely on the main soluble component of the SVC, synapsin, as their key modulator (Yu et al., 2021), indicating that the SVC may control synaptic stability by modulating BDNF release. Incidentally, this provides another synaptic role for the SVC, that of controller of the distribution and availability of dense-core vesicles. Additional details on their distribution are provided in Robinson et al. (2016).

Accumulating evidence also supports an important role for the extracellular matrix (ECM) in regulating synaptic function, which may be particularly relevant for synapse stabilization via interactions with both the pre- and post-synapse (Dityatev et al., 2010; Ferrer-Ferrer & Dityatev, 2018). A prominent example of such interplay at the pre-synaptic site is the modulation of SV release by reelin. It has been demonstrated that adding reelin to hippocampal cultured neurons led to an increase in spontaneous neuro-transmitter release, possibly through interactions of both

very-low-density-lipoprotein receptor and apolipoprotein E receptor 2 with reelin (Bal et al., 2013). Additionally, it was reported that the action of reelin depends on the function of the SV protein identified as vesicle-associated membrane protein 7, a molecule considered to be a marker of the reserve pool of vesicles (Hua et al., 2011), indicating that ECM dynamics interact with SV functionality (albeit it is not clear whether the opposite would also be true).

An additional example of presynaptic modulation by ECM molecules is the case of laminin, a vital player in the organization of the presynaptic release apparatus. It has been demonstrated that, in the adult retina, a deficiency in laminin $\beta2$ resulted in the perturbation of the arrangement of various presynaptic components, although their expression levels remained constant (Hunter et al., 2019). This mechanism is presumably implemented via interactions of laminin $\beta2$ with the extracellular domain of molecules comprising the presynaptic release machinery, a phenomenon that has already been reported at the neuromuscular junction synapses (Nishimune et al., 2004). Interestingly, a direct interaction of laminin $\alpha5$ subunit with SV2, known for its role in the process of SV priming, was also reported (Chang & Südhof, 2009; Son et al., 2000). Furthermore, indirect interactions of laminin with the synapse via ECM receptors such as integrins need to be taken into consideration. The laminin $\alpha5$ binding partner, integrin $\beta1$, has been reported to be present at the presynaptic site in hippocampal neurons (Mortillo et al., 2012) and a neuron-specific deficiency in integrin $\beta1$ led to alterations in synaptic responses, consistent with decreased mobilization of vesicles from the reserve pool (Huang et al., 2006). Finally, it is becoming increasingly clear that the presence of the ECM may have a role in synaptic plasticity, by providing an anchor point for postsynaptic actin dynamics (Bonilla-Quintana & Rangamani, 2024).

Overall, these findings provide evidence on the interaction between the SVC and the ECM. However, it remains unclear what the main regulator of this process is. Although disruption of the ECM resulted in altered synaptic activity, the opposite reaction has not been a focus of substantial research. Indeed, proteomics data demonstrated that various ECM binding partners, including integrins, CD44, NCAM and Igsf8, can be found in SVs (Taoufiq et al., 2020), suggesting that SVs play an important role in the initial organization of the ECM at the synapse. Although speculative, this hypothesis should be tested in the future.

Another mechanism of synapse stabilization involves glial cells. Not only microglia, but also astrocytes are known to actively participate in synaptic pruning via direct phagocytosis (Chung et al., 2013) or through various signalling cascades (Vainchtein et al., 2018; Yang et al., 2016). Astrocytes are involved in tight contacts with synapses and respond to neurotransmitter release (Araque et al., 1999; Bernardinelli et al., 2014), providing an additional pathway by which the activity of the SVs can affect synaptic stability.

Although synaptic stabilization is crucial for both cognitive and motor functions and its loss is generally associated with neurodegenerative disorders (Wishart et al., 2006), excessive stabilization may also lead to pathological conditions. The majority of synaptic pruning events occur during early stages of neurodevelopment, when many synapses form and are then quickly removed prior to full maturation. It has been suggested that failure to remove these 'unneeded' synapses and the resulting hyperconnectivity might underlie delayed brain development and psychoneurological symptoms of psychiatric disorders (Xie et al., 2023).

In addition, the stabilization of synaptic components, such as organelles and protein complexes, may be an important cue for pathological developments. Most synaptic proteins turn over during an interval of a few days to a few weeks (Fornasiero et al., 2018), and this process is necessary for synapse function. For example, a failure to degrade results in the formation of long-lasting aSyn fibrils, ultimately resulting in the formation of pathognomonic inclusions known as Lewy bodies (Brás et al., 2020; Calabresi et al., 2023; Nordengen & Morland, 2024). The incorporation of aSyn into the SVC might serve a protective role by maintaining it in a liquid-condensed state, as opposed to solid (stable) aggregates, without reducing its concentration (Brodin et al., 2022). This hints towards possible additional roles of the SVC in preventing neurodegeneration by reducing excessive molecular stabilization.

## Conclusions

The numerous connections and relationships between the SVC and other synaptic components, both on the pre- and postsynaptic side, indicate that much of the known synaptic biology could be regulated by the SVC at least in part, either directly or indirectly. This regulatory role of the SVC may not only depend on the protein interactions discussed above, but also equally involve lipid-dependent interactions and signalling. Apart from modulating excocytotic and endocytotic processes (Man et al., 2021), the lipid composition of SVs (Binotti et al., 2021) probably supports synaptic metastasis more generally. To this end, a transcriptional feedback loop between lipid biosynthesis and the synaptic vesicle pool has already been reported (Tsai et al., 2019). At the same time, the role of vesicular trafficking and lipid metabolism has been acknowledged in synucleinopathies (Fanning et al., 2021) and more generally in neurodegeneration (Beal et al., 2020; Yoon et al., 2022). Nevertheless, it is crucial to acknowledge that not all synaptic processes can

be regulated by synaptic vesicle pools, and exceptions will exist, linked, for example, to synapse heterogeneity (e.g. profound differences between glutamatergic and GABAergic synapses, despite relatively minor differences in their proteomes) (Grønborg et al., 2010). Other elements remain as open questions, such as, for example, the recruitment of regulatory receptors. These receptors wield considerable influence over synaptic transmission, by regulating the activity of neural networks, thereby accommodating various environmental conditions. How are their localization and behaviour regulated? Are they equally distributed through neurons, akin to ACh receptors prior to synapse formation? What initiates and drives their re-localization to synapses?

Answers to these questions might come from future work on improving the tools for synaptic analysis, from improved imaging to superior data analysis. Recent years have already seen a significant improvement in these fields, achieving Ångstrom-scale resolution with light microscopy, which, when combined with improved labelling strategies and novel deep learning-based data processing techniques, can provide unprecedented insight into synaptic biology, possibly by synergizing with tools from the cryo-electron tomography (cryo-ET) family (Ebrahimi et al., 2023; Jong-Bolm et al., 2023; Ostersehlt et al., 2022; Sahl et al., 2023; Shaib et al., 2023; Unterauer et al., 2024). Such enhanced visualization methodologies and novel data analysis approaches promise to further facilitate more exhaustive and precise investigations into the mechanisms underpinning synaptic transmission and plasticity.

Moreover, novel registration and deep learning-based entity detection algorithms will pave the way to better detect and localize synaptic proteins and organelles, from different types of samples, spanning different resolution ranges (Kylies et al., 2023). One of the most promising directions is the automated reconstruction of synaptic structures in volume electron microscopy and ET for a detailed understanding of synapse morphology. Deep learning-based methods have been used very successfully for related tasks, such as mitochondrion segmentation (Conrad & Narayan, 2023) and segmentation of cellular ultrastructure (Heinrich et al., 2021). Some methods already apply this approach to synaptic vesicles (Imbrosci et al., 2022; Khosrozadeh et al., 2024) and further improvements, in particular, training on larger and more diverse datasets and for more structures, should enable automatic synapse reconstruction. Similarly, deep learning-based methods are now established for analyzing macromolecular assemblies *in situ* based on cryo-ET (de Teresa-Trueba et al., 2023; Rice et al., 2023). Adapting these methods to the synapse by extending them to membrane-bound proteins and protein complexes would enable a much better understanding of the synapse's molecular organization. This extension appears to be feasible as a result of precise computational predictions for protein structures (Jumper et al., 2021) and protein complexes (Abramson et al., 2024) that can support the *in situ* analysis, both for cryo-ET and high-resolution light microscopy.

Additionally, new minimal invasive tools to follow synaptic protein in living neurons, new molecular sensors for detecting changes in synaptopathologies and optogenetic manipulations of the SVC have been developed (Gerdes et al., 2020; Queiroz Zetune Villa Real et al., 2023; Rimbault et al., 2024; Rivera et al., 2023; Vettkötter et al., 2022; Won et al., 2022). Such enhanced labelling, tracking and visualization methodologies combined with novel data analysis approaches promise to facilitate further exhaustive and precise investigations into the mechanisms underpinning synaptic transmission and plasticity. Moreover, advancements in chemical probe design will make it possible to interrogate and investigate neuronal structure and signalling in novel ways. For example, light-sensitive probes should allow for increasingly complex manipulation of neuronal firing and the signalling of synaptic receptors (DiFrancesco et al., 2020; Garrido-Charles et al., 2022; Kramer & Rajappa, 2022). The diversification of chemical labelling reactions made it possible to visualize native receptors *in vivo* using, for example, ligand directed (Nonaka et al., 2024; Ojima et al., 2021; Wakayama et al., 2017) functionalization or bioorthogonal labelling with non-canonical amino acids (Bessa-Neto et al., 2021; Nikić-Spiegel, 2020). Moreover, (opto)chemical probes that combine labelling with erasing strategies have already been validating the benefits of improvement in probe design (Hayashi-Takagi et al., 2015; Mougios et al., 2024) and will continue to contribute significantly to the advancement in understanding synaptic structures and function.

Overall, we conclude that, although multiple synapse-regulation features could be assigned to the SVC, current technology improvements will probably enhance this list, towards aspects that we could not yet cover, ranging from synaptic tuning by rare molecules (e.g. neuromodulatory receptors) to synaptic protein translation.

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

## Additional information

### Competing interests

The authors declare that they have no competing interests.

### Author contributions

All of the authors contributed to writing the manuscript. All of the authors have read and approved the final version of the manuscript submitted for publication.

### Funding

This work was supported by the German Research Foundation (Deutsche Forschungsgemeinschaft, DFG) through grants SFB1286/A01 (to BC), SFB1286/A02 (to TS), SFB1286/A03 (to SOR), SFB1286/A04 (to CW), SFB1286/A05 (to SJ), SFB1286/A06 (to PR), SFB1286/A07 (to ED), SFB1286/A08/A10 (to HU), SFB1286/A09 (to NB and MT), SFB1286/A11 (to NL), SFB1286/A12 (to RF-B), SFB1286/B02 (to SKo), SFB1286/B04 (to CS), SFB1286/B05 and SFB1286/C8 (to TM), SFB1286/B06 (to AF), SFB1286/B08 (to TO), SFB1286/B10 (to DM), SFB1286/B11 (to OS), SFB1286/C01 (to CT), SFB1286/C02 (to FW), SFB1286/C03 (to MF), SFB1286/C05 (to SKl), SFB1286/C06 (to MM), SFB1286/C09 (to VP), SFB1286/C10 (to CP), SFB1286/Z02 (to SB) and SFB1286/Z04 (to FO).

### Acknowledgements

### Keywords

synapse, synapse formation, synaptic plasticity, synaptic vesicle cluster, vesicle

## Supporting information

Additional supporting information can be found online in the Supporting Information section at the end of the HTML view of the article. Supporting information files available:

**Peer Review History**

