## [Peer Review History · The Journal of Physiology]

The synaptic vesicle cluster as a controller of pre- and postsynaptic structure and function

Sofia Reshetniak, Cristian A Bogaciu, Stefan Bonn, Nils Brose, Benjamin H. Cooper, Elisa D'Este, Michael Fauth, Rubén Fernández-Busnadiego, Maksims Fiosins, André Fischer, Svilen V. Georgiev, Stefan Jakobs, Stefan Klumpp, Sarah Köster, Felix Lange, Noa Lipstein, Victor Macarrón Palacios, Dragomir Milovanovic, Tobias Moser, Marcus Müller, Felipe Opazo, Tiago F. Outeiro, Constantin Pape, Viola Priesemann, Peter Rehling, Tim Salditt, Oliver Schlüter, Nadja Simeth-Crespi, Claudia Steinem, Tatjana Tchumatchenko, Christian Tetzlaff, Marilyn Tirard, Henning Urlaub, Carolin Wichmann, Fred Wolf, and Silvio O Rizzoli

DOI: 10.1113/JP286400

Corresponding author(s): Silvio Rizzoli (srizzol@gwdg.de)

The following individual(s) involved in review of this submission have agreed to reveal their identity: Shigeo Takamori (Referee #2)

Review Timeline:	Submission Date:	12-Jun-2024
	Editorial Decision:	12-Jul-2024
	Revision Received:	19-Aug-2024
	Accepted:	11-Sep-2024

Senior Editor: Laura Bennet

Reviewing Editor: Samuel Young

Transaction Report:

Dear Professor Rizzoli,

Re: JP-TR-2024-286400 "The synaptic vesicle cluster as a controller of pre- and postsynaptic structure and function" by Sofiia Reshetniak, Cristian A Bogaciu, Stefan Bonn, Nils Brose, Benjamin Cooper, Elisa D'Este, Michael Fauth, Rubén Fernández-Busnadiego, Maksims Fiosins, André Fischer, Svilen V. Georgiev, Stefan Jakobs, Stefan Klumpp, Sarah Köster, Felix Lange, Noa Lipstein, Victor Macarrón Palacios, Dragomir Milovanovic, Tobias Moser, Marcus Müller, Felipe Opazo, Tiago Outeiro, Constantin Pape, Viola Priesemann, Peter Rehling, Tim Salditt, Oliver Schlüter, Nadja Simeth-Crespi, Claudia Steinem, Tatjana Tchumatchenko, Christian Tetzlaff, Marilyn Tirard, Henning Urlaub, Carolin Wichmann, Fred Wolf, and Silvio O Rizzoli

Thank you for submitting your manuscript to The Journal of Physiology. It has been assessed by a Reviewing Editor and by 2 expert referees and we are pleased to tell you that it is acceptable for publication following satisfactory revision.

ABSTRACT FIGURES: Authors may use The Journal's premium BioRender account to create/redraw their Abstract Figures (and any other suitable schematic figure). Information on how to access this account is here: <https://physoc.onlinelibrary.wiley.com/journal/14697793/biorender-access>.

REVISION CHECKLIST: Upload a full Response to Referees file. To create your 'Response to Referees' copy all the reports, including any comments from the Senior and Reviewing Editors, into a Microsoft Word, or similar, file and respond to each point, using font or background colour to distinguish comments and responses and upload as the required file type.

We look forward to receiving your revised submission.

Yours sincerely,

Laura Bennet
Senior Editor
The Journal of Physiology

EDITOR COMMENTS

Reviewing Editor:

This broad multi-author review centers on the known and potential roles of synaptic vesicle clusters in the regulation of synapse function. Both reviewers found the review very thorough and highly influential for the field of synaptic biology. However, both reviewers have concerns. In particular, there appears to be mixing of what is actually a review that is based on what is currently known in the field and what is a perspective/speculation that is interpretations of current findings. This mixing leads to confusion. Furthermore, both reviewers have commented on the section on SVCs on synapse initiation that it is distantly related to the review topic. Taken together, to address these concerns the authors should do a careful revision of the text in response to the reviewers positive criticism. It may be beneficial for the authors have a separate speculation section in the review to help clarify what is known vs what is speculation and are unsolved open questions for future directions in the field.

Please also see 'Required Items' below.

REFEREE COMMENTS

Referee #1:

Reshetniak et al in this broad multi-author review discuss the known and putative roles of the synaptic vesicle cluster (SVC) in the regulation of pre- and postsynaptic functions. The authors summarize current concepts regarding the molecular organization of the SVC and the possibility of differential expression of SVC encoding genes in different types of neurons. They then move on to discuss known and somewhat more hypothetical roles of SVs and other factors contained within the SVC in synapse initiation, maturation, and function before moving to the interrelationship between the SVC and various other organelles such as mitochondria, the ER and, finally, to synaptic plasticity, synapse stabilization and loss including the possible function of the extracellular matrix.

I regard this review as a valuable resource to researchers entering the field of presynaptic biology as well as for experts seeking to better understand the connection between the SVC and synapse function. The topics covered are very broad and many of the conclusions drawn appear quite general. This sometime comes at the expense of detail and precision as further explained in my specific comments.

1. At least to this referee it is not always clear which aspects of the review reflect known fact supported by experimental data and which represent a hypothesis to be tested or, speculation by the authors. For example, in the chapter on synapse initiation it is suggested that axon growth is fueled by a mixed type of vesicles comprising various kinds of presynaptic cargo including SV proteins and cell adhesion molecules. At least to this referee, there is no experimental evidence that the machineries for axon growth and presynapse formation are identical or even closely linked. Recent data on fly and human neurons carrying mutations in liprin-a proteins or lacking anterograde axonal motors or adaptors (e.g. Marco de la Cruz et al 2024; Rizalar et al 2023; Vukoja et al 2018) important for presynaptic delivery of SV and AZ proteins did not reveal defects in axon growth. Hence, whether delivery of axonal precursors of SVs contributes to axon growth remains an open question. Likewise, at least to this referee it is unknown whether axonal precursor vesicles mature into SVs by recycling as stated on p. 10.

2. While I appreciate the breadth of the review, some aspects covered appear to be related only distantly to the main topic of the review. For example, whether and how the SVC contribute to or controls the presence or absence of synaptic mitochondria is unclear. The speculation that competition between mitochondria and SV precursors for motor proteins, at

least according to this review, does not seem to be supported by data. Instead, recent studies have demonstrated that mitochondria and SV precursors are transported along axons largely independently of each other during development. Although it cannot be ruled out that competition may exist in defined neurons or synapse subtypes, the hypothesis seems somewhat far-fetched.

3. p.17/ mRNA: There is accumulating evidence for axonal transport of mRNA encoding mitochondrial and ribosomal proteins via piggyback riding of RNA granules on axonally transported lysosomes (and possibly on SV precursors). These works should be quoted in this context (De Pace et al. Nat Neurosci. 2024; Cioni et al Cell 2019; Liao et al Cell 2019).

4. p.18: The statement: Third, the endoplasmic reticulum is known to form numerous membrane contact sites with various organelles (Wu et al., 2017), including organelles containing Rab7 and phosphatidylinositol 3-phosphate (Raiborg et al., 2015). Both of these molecules are abundant in SVs, therefore the SVs might directly affect the ER via these contact sites.

At least to this referee there is no evidence that SVs contain PI(3)P.

5. Given the breadth of topics covered the authors may consider adding further illustrations that cover the core messages of the review. I leave this to the discretion of the authors.

Referee #2:

This review posits that the synaptic vesicle cluster (SVC) operates as a co-factor buffering system and explores its potential roles beyond SV recycling. This perspective is highly intriguing and represents an influential review shaping future research directions. It is a compelling read, indeed. Therefore, this reviewer strongly recommends accepting the manuscript. However, prior to final acceptance, reconsideration and revision of the following points are recommended. Additionally, minor typos and citation errors were noted, necessitating correction.

1. The initial chapter on Synapse Initiation, which discusses stages prior to SVC formation seems redundant and deviates from the review's primary focus. Consider shortening or omitting it to enhance the main topic.

2. Throughout the review, the discussion appears to conflate the functions of the SVC with the effects of other factors influencing the SVC (e.g., synaptic mitochondria, coincidental effects on the postsynaptic side). This distinction should be clarified wherever possible.

3. The conclusion proposes future research strategies, highlighting the necessity for advanced imaging techniques and data analysis to elucidate SVC functions. Shouldn't recent advancements in light-manipulation methods for SVC (e.g., Won et al., Neuron 110(3):423-435, 2022; Vettkötter et al., Nat Commun 13(1):7827, 2022) also be introduced?

Minor corrections:

1. Abstract: "extend the functional involvement of the SVC in synapse formation..."

2. "Camilli PD" should be corrected to "De Camilli, P."

3. P. 9: (although this reviewer recommends to delete or shorten this part) 'they rely on scaffold components to form the SVC'

4. Page 9: Correct reference from "Chen et al., 2023" to "Chen et al., 2020."

5. Page 9: "Alfken et al., 2024" is missing from the reference list.

6. Page 21: "Binotti et al., 2021" does not appear in the reference list.

7. Page 26: Remove extra phrases from "Chen et al., 2023."

8. Page 35: Correct the duplication of the Miki et al. paper.

REQUIRED ITEMS

- Please include an Abstract Figure file, as well as the Figure Legend text within the main article file. The Abstract Figure is a piece of artwork designed to give readers an immediate understanding of the Review Article and should summarise the main conclusions. If possible, the image should be easily 'readable' from left to right or top to bottom. It should show the physiological relevance of the Review so readers can assess the importance and content of the article. Abstract Figures should not merely recapitulate other figures in the Review. Please try to keep the diagram as simple as possible and without superfluous information that may distract from the main conclusion of the Review. Abstract Figures must be provided by authors no later than the revised manuscript stage and should be uploaded as a separate file during online submission labelled as File Type 'Abstract Figure'. Please ensure that you include the figure legend in the main article file. All Abstract Figures will be sent to a professional illustrator for redrawing and you may be asked to approve the redrawn figure before your paper is accepted.

- Please upload separate high quality figure files via the submission form.

- Please ensure that the Article File you upload is a Word file.

END OF COMMENTS

Confidential Review

12-Jun-2024

Reply to the Referee comments

Reviewing Editor:

This broad multi-author review centers on the known and potential roles of synaptic vesicle clusters in the regulation of synapse function. Both reviewers found the review very thorough and highly influential for the field of synaptic biology. However, both reviewers have concerns. In particular, there appears to be mixing of what is actually a review that is based on what is currently known in the field and what is a perspective/speculation that is interpretations of current findings. This mixing leads to confusion. Furthermore, both reviewers have commented on the section on SVCs on synapse initiation that it is distantly related to the review topic. Taken together, to address these concerns the authors should do a careful revision of the text in response to the reviewers positive criticism. It may be beneficial for the authors have a separate speculation section in the review to help clarify what is known vs what is speculation and are unsolved open questions for future directions in the field.

We thank the Editor for the suggestions.

To address the different comments, we added a table (Table 1), in which we explain what is currently known, and what our speculations are.

In addition, we shortened the section on synapse initiation substantially.

Referee #1:

Reshetniak et al in this broad multi-author review discuss the known and putative roles of the synaptic vesicle cluster (SVC) in the regulation of pre- and postsynaptic functions. The authors summarize current concepts regarding the molecular organization of the SVC and the possibility of differential expression of SVC encoding genes in different types of neurons. They then move on to discuss known and somewhat more hypothetical roles of SVs and other factors contained within the SVC in synapse initiation, maturation, and function before moving to the interrelationship between the SVC and various other organelles such as mitochondria, the ER and, finally, to synaptic plasticity, synapse stabilization and loss including the possible function of the extracellular matrix.

We thank the Referee for the comments.

I regard this review as a valuable resource to researchers entering the field of presynaptic biology as well as for experts seeking to better understand the connection between the SVC and synapse function. The topics covered are very broad and many of the conclusions drawn appear quite general. This sometime comes at the expense of detail and precision as further explained in my specific comments.

1. At least to this referee it is not always clear which aspects of the review reflect known fact supported by experimental data and which represent a hypothesis to be tested or, speculation by the authors. For example, in the chapter on synapse initiation it is suggested that axon growth is fueled by a mixed type of vesicles comprising various kinds of presynaptic cargo including SV proteins and cell adhesion molecules. At least to this referee, there is no experimental evidence that the machineries for axon growth and presynapse formation are identical or even closely linked. Recent data on fly and human neurons carrying mutations in liprin-a proteins or lacking anterograde axonal motors or adaptors (e.g. Marco de la Cruz et al 2024; Rizalar et al 2023; Vukoja et al 2018) important for presynaptic delivery of SV and AZ proteins did not reveal defects in axon growth. Hence, whether delivery of axonal precursors of SVs contributes to axon growth remains an open question. Likewise, at least to this referee it is unknown whether axonal precursor vesicles mature into SVs by recycling as stated on p.

10.

As discussed in the respective section, vesicles used for axonal growth contain *bona fide* SV proteins, thus exhibiting the characteristics of precursor vesicles, leading to the hypothesis we expressed.

We have now shortened the respective section, and we also revised it, including the works indicated by the Referee, clearly indicating the steps on which we only speculate, and the ones that have been already proven by numerous works.

2. While I appreciate the breadth of the review, some aspects covered appear to be related only distantly to the main topic of the review. For example, whether and how the SVC contribute to or controls the presence or absence of synaptic mitochondria is unclear. The speculation that competition between mitochondria and SV precursors for motor proteins, at least according to this review, does not seem to be supported by data. Instead, recent studies have demonstrated that mitochondria and SV precursors are transported along axons largely independently of each other during development. Although it cannot be ruled out that competition may exist in defined neurons or synapse subtypes, the hypothesis seems somewhat far-fetched.

We now indicate that the respective hypothesis is largely speculative.

3. p.17/ mRNA: There is accumulating evidence for axonal transport of mRNA encoding mitochondrial and ribosomal proteins via piggybac riding of RNA granules on axonally transported lysosomes (and possibly on SV precursors). These works should be quoted in this context (De Pace et al. Nat Neurosci. 2024; Cioni et al Cell 2019; Liao et al Cell 2019).

We now include these works.

4. p.18: The statement: Third, the endoplasmic reticulum is known to form numerous membrane contact sites with various organelles (Wu et al., 2017), including organelles containing Rab7 and phosphatidylinositol 3-phosphate (Raiborg et al., 2015). Both of these molecules are abundant in SVs, therefore the SVs might directly affect the ER via these contact sites. At least to this referee there is no evidence that SVs contain PI(3)P.

It is thought that the presence of PI(3)P is necessary for the endosomal processing of SVs, which is an important step in their recycling (e.g. Binotti et al., 2021, doi: 10.1016/j.abb.2021.108966; Lauwers et al., 2016; doi: 10.1016/j.neuron.2016.02.033). To our knowledge, the first investigation of the presence of this lipid on vesicles dates more than 20 years ago (Wucherpfennig et al., 2003; doi: 10.1083/jcb.200211087), employing FYVE domains to detect the presence of PI(3)P in immuno-electron microscopy. While most FYVE domains are found on endosome-like cisternae, 8% are found on SVs. We now mention these works in the respective section.

5. Given the breadth of topics covered the authors may consider adding further illustrations that cover the core messages of the review. I leave this to the discretion of the authors.

We now added a new Table 1 which summarizes the most important effects we discuss, and categorizes them based on whether they have been demonstrated experimentally, are hypothesized based on available data, and whether they stem from direct or indirect action of the SVC cluster.

Referee #2:

This review posits that the synaptic vesicle cluster (SVC) operates as a co-factor buffering system and explores its potential roles beyond SV recycling. This perspective is highly intriguing and represents an influential review shaping future research directions. It is a compelling read, indeed. Therefore, this reviewer strongly recommends accepting the manuscript. However, prior to final acceptance, reconsideration and revision of the following points are recommended. Additionally, minor typos and citation errors were noted, necessitating correction.

We thank the Referee for the comments.

1. The initial chapter on Synapse Initiation, which discusses stages prior to SVC formation seems redundant and deviates from the review's primary focus. Consider shortening or omitting it to enhance the main topic.

We have now shortened this section by 35%.

2. Throughout the review, the discussion appears to conflate the functions of the SVC with the effects of other factors influencing the SVC (e.g., synaptic mitochondria, coincidental effects on the postsynaptic side). This distinction should be clarified wherever possible.

We have now tried to disentangle these effects by including a new Table 1 that indicates whether the discussed pathways are directly influenced by the SVC or have an indirect connection.

3. The conclusion proposes future research strategies, highlighting the necessity for advanced imaging techniques and data analysis to elucidate SVC functions. Shouldn't recent advancements in lightmanipulation methods for SVC (e.g., Won et al., Neuron 110(3):423-435, 2022; Vettkötter et al., Nat Commun 13(1):7827, 2022) also be introduced?

We have now added this important idea.

Minor corrections:

- 1. Abstract: "extend the functional involvement of the SVC in synapse formation..."*
- 2. "Camilli PD" should be corrected to "De Camilli, P."*
- 3. P. 9: (although this reviewer recommends to delete or shorten this part) 'they rely on scaffold components to form the SVC'*
- 4. Page 9: Correct reference from "Chen et al., 2023" to "Chen et al., 2020."*
- 5. Page 9: "Alfken et al., 2024" is missing from the reference list.*
- 6. Page 21: "Binotti et al., 2021" does not appear in the reference list.*
- 7. Page 26: Remove extra phrases from "Chen et al., 2023."*
- 8. Page 35: Correct the duplication of the Miki et al. paper.*

We have made these corrections.

Dear Professor Rizzoli,

Re: JP-TR-2024-286400R1 "The synaptic vesicle cluster as a controller of pre- and postsynaptic structure and function" by Sofiia Reshetniak, Cristian A Bogaciu, Stefan Bonn, Nils Brose, Benjamin H. Cooper, Elisa D'Este, Michael Fauth, Rubén Fernández-Busnadiego, Maksims Fiosins, André Fischer, Svilen V. Georgiev, Stefan Jakobs, Stefan Klumpp, Sarah Köster, Felix Lange, Noa Lipstein, Victor Macarrón Palacios, Dragomir Milovanovic, Tobias Moser, Marcus Müller, Felipe Opazo, Tiago F. Outeiro, Constantin Pape, Viola Priesemann, Peter Rehling, Tim Salditt, Oliver Schlüter, Nadja Simeth-Crespi, Claudia Steinem, Tatjana Tchumatchenko, Christian Tetzlaff, Marilyn Tirard, Henning Urlaub, Carolin Wichmann, Fred Wolf, and Silvio O Rizzoli

We are pleased to tell you that your paper has been accepted for publication in The Journal of Physiology.

Authors should note that it is too late at this point to offer corrections prior to proofing. Major corrections at proof stage, such as changes to figures, will be referred to the Editors for approval before they can be incorporated. Only minor changes, such as to style and consistency, should be made at proof stage. Changes that need to be made after proof stage will usually require a formal correction notice.

Yours sincerely,

Laura Bennet
Senior Editor
The Journal of Physiology

P.S. - You can help your research get the attention it deserves! Check out Wiley's free Promotion Guide for best-practice recommendations for promoting your work at www.wileyauthors.com/eoo/guide. You can learn more about Wiley Editing Services which offers professional video, design, and writing services to create shareable video abstracts, infographics, conference posters, lay summaries, and research news stories for your research at www.wileyauthors.com/eoo/promotion.

IMPORTANT NOTICE ABOUT OPEN ACCESS: To assist authors whose funding agencies mandate public access to published research findings sooner than 12 months after publication, The Journal of Physiology allows authors to pay an Open Access (OA) fee to have their papers made freely available immediately on publication.

You can check if your funder or institution has a Wiley Open Access Account here: <https://authorservices.wiley.com/author-resources/Journal-Authors/licensing-and-open-access/open-access/author-compliance-tool.html>.

EDITOR COMMENTS

Reviewing Editor:

The authors have responded to all previous critiques. There are no further concerns.

REFEREE COMMENTS

Referee #1:

My previous comments have all been addressed and I recommend publication of this nice review.

1st Confidential Review

19-Aug-2024